
# Analytic Solutions for Long's Equation and its Generalization

Mayer Humi

Department of Mathematical Sciences

Worcester Polytechnic Institute

100 Institute Road

Worcester, MA 01609

## Abstract

Two dimensional, steady state, stratified, isothermal, atmospheric flow over topography is governed by Long's equation. Numerical solutions of this equation were derived and used by several authors. In particular these solutions were applied extensively to analyze the experimental observations of gravity waves. In the first part of this paper we derive an extension of this equation to non-isothermal flows. Then we devise a transformation that simplifies this equation. We show that this simplified equation admits solitonic type solutions in addition to regular gravity waves. These new analytical solutions provide insights about the propagation and amplitude of gravity waves over topography.

PACS 92.60.Gn, 92.60.Dj, 02.30.Ik





# 1   Introduction

Two dimensional steady state flow of isothermal, incompressible stratified fluid over topography is modeled by Long's equation [Long 1953, Long 1954, Long 1955, Long 1959]. Numerical solutions of this equation for base flow without shear over simple terrain, which consists of one hill, were derived and analyzed in the literature by several authors.[Drazin 1961, Yih 1967, Drazin and Moore 1967, Lily and Klemp 1979, Smith 1980, Peltier and Clark 1983, Smith 1989, Durran 1992,Smith and Kruse 2017].

In these studies it was usual to approximate the Brunt-Väisälä frequency by a constant or a step function. In addition two physical parameters which control the stratification and dispersive effects of the atmosphere were set to zero. Under these approximations, one of the leading second order derivatives in Long's equation drop out. Moreover the equation become linear (the nonlinear terms disappear). In this singular limit Long's equation reduces to that of a linear harmonic oscillator over the computational domain. The impact of these approximations on the validity of the solution was analyzed in depth in the literature [Smith 1980, Peltier and Clark 1983, Smith 1989]. These studies demonstrated that these approximations set limits on the physical applicability of these solutions.

Solutions of Long's equation were used also as a framework for the examination and study of experimental data on gravity waves. [Shutts et al 1988, Shutts et al 1994, Fritts and Alexander 2003, Jumper et al 2004, Vernin et al 2007, Richter et al 2010, Geller et al 2013]. In all of these studies it was assumed that the base flow is shearless. However this assumption is incorrect, in general, and is not justified by the experimental data. (For a comprehensive list of references see [Yih 1980,Baines 1995,Nappo 2012]).

A new method to derive analytic solutions of Long's equation was initiated by the present author in [Humi 2004, Humi 2007, Humi 2009, Humi 2010 ,Humi 2015]. It was demonstrated that Long's equation can be simplified for shearless base flow with mild assumptions on the nonlinear terms. In this framework we were able to identify the "slow variable" in Long's equation. This





variable controls the emergence of nonlinear oscillations in this equation. In addition we proved the existence of self similar solutions and derived a formula for the attenuation of the gravity waves amplitude with height. These results follow from the general properties of Long's equation and the nonlinear terms present in this equation.

We considered the effect that shear in the base flow has on the generation of gravity waves and their amplitude in [Humi 2006]. A new form of Long's equation in which the stream function is replaced by by the atmospheric density was derived in [Humi 2007]. Finally a generalization of Long's equation to time dependent flows appeared in [Humi 2015].

It obvious however that atmospheric flows over topography are not isothermal in general (see [Miglietta and Rotunno 2014, Richter et al 2010, Smith and Kruse,2017] and their bibliography). With this motivation we derive, in the first part of this paper, an extension of this equation to include non-isothermal flows with free convection.

In the second part of the paper we devise a new transformation on Long's equation (isothermal or not) that yields new analytic solutions for the perturbation from the base flow (under mild approximations). In particular we demonstrate that there exist "solitonic type solutions" to this equation in addition regular gravity waves. We derive also an expression which relates the change of the amplitude of the gravity waves as a function of height.

The plan of the paper is as follows: In the first part of Sec. 2 we presents an overview of the derivation of the isothermal Long's equation. In the second part we derive the corresponding Long's equation for flows with free convection. In Sec. 3 we introduce a transformation which (essentially) linearizes the equation for the perturbation from the base flow. Sec 4 discusses the application of this transformation to a flow with shear. We summarize with some conclusions in Section 5.

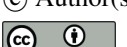



# 2 Derivation of Long's Equation

In the first part of this section we provide a short overview of the (classical) isothermal Long's equation and in the second part we generalize this equation to include free convection.

## 2.1 Isothermal Long's Equation

In two dimensions $(x, z)$ the flow of a steady isothermal, inviscid and incompressible stratified fluid is modeled by the following equations:

$$u_x + w_z = 0 \tag{2.1}$$

$$u\rho_x + w\rho_z = 0 \tag{2.2}$$

$$\rho(uu_x + wu_z) = -p_x \tag{2.3}$$

$$\rho(uw_x + ww_z) = -p_z - \rho g. \tag{2.4}$$

In these equations subscripts denote differentiation with respect to the subscripted variable, $\mathbf{u} = (u, w)$ is the fluid velocity, $p$ denotes the pressure, $\rho$ denotes the density and $g$ is the acceleration of gravity,

To non-dimensionalize (2.1)-(2.4) we introduce the following scaled variables,

$$
\begin{aligned}
\bar{x} &= \frac{x}{L}, \ \ \bar{z} = \frac{N_0}{U_0}z, \ \ \bar{u} = \frac{u}{U_0}, \ \ \bar{w} = \frac{LN_0}{U_0^2}w \\
\bar{\rho} &= \frac{\rho}{\bar{\rho}_0}, \ \ \bar{p} = \frac{N_0}{gU_0\bar{\rho}_0}p,
\end{aligned}
\tag{2.5}
$$

In these equations $L$ represents a characteristic length, and $U_0$ is the free stream velocity, and $\bar{\rho}_0$ is the averaged base density which is considered to be a constant. $N_0^2$ represents an averaged value of the Brunt-Väisälä frequency which is defined as

$$N^2 = -\frac{g}{\rho_0}\frac{d\rho_0}{dz} \tag{2.6}$$



where $\rho_0(z)$ is the base density.

Using these new variables (2.1)-(2.4) take the following form (the bars were dropped for brevity)

$$u_x + w_z = 0 \tag{2.7}$$

$$u\rho_x + w\rho_z = 0 \tag{2.8}$$

$$\beta\rho(uu_x + wu_z) = -p_x \tag{2.9}$$

$$\beta\rho(uw_x + ww_z) = -\mu^{-2}(p_z + \rho). \tag{2.10}$$

Where,

$$\mu = \frac{U_0}{N_0 L}. \tag{2.11}$$

$$\beta = \frac{N_0 U_0}{g}, \tag{2.12}$$

In these equations $\mu$ is the long wave parameter which controls dispersive effects or equivalently the deviation from the hydrostatic approximation. When $\mu = 0$ the hydrostatic approximation is fully satisfied [Smith 1980,Smith 1989]. The coefficient $\beta$ is the "Boussinesq parameter" [Baines 1995,Nappo 2012], which controls stratification effects (assuming $U_0 \neq 0$)

Equation (2.7) implies that it is possible to introduce a stream function $\psi$ so that

$$u = \psi_z, \quad w = -\psi_x . \tag{2.13}$$

Using this definition of $\psi$ it is possible to rewrite (2.8) as

$$J\{\rho, \psi\} = 0. \tag{2.14}$$

The symbol $J(f, g)$ is defined for any two smooth functions $f, g$ as

$$J\{f, g\} = \frac{\partial f}{\partial x}\frac{\partial g}{\partial z} - \frac{\partial f}{\partial z}\frac{\partial g}{\partial x} \tag{2.15}$$

It is easy to show that when $J(f, g) = 0$ it is possible to express each of these functions in terms of the other [Yih 1980]. It follows then from (2.14) that the functions $\rho, \psi$ are dependent on each other. This means that one can express $\rho$ as $\rho(\psi)$ or $\psi$ as $\psi(\rho)$.





Using (2.13 one can rewrite the momentum equations (2.9), (2.10) in terms of $\psi$.

$$\beta\rho(\psi_z\psi_{zx} - \psi_x\psi_{zz}) = -p_x \tag{2.16}$$

$$\beta\rho(-\psi_z\psi_{xx} + \psi_x\psi_{xz}) = -\mu^{-2}(p_z + \rho) \tag{2.17}$$

To eliminate $p$ from (2.16), (2.17) we multiply (2.17) by $\mu^2$ and then differentiate (2.16), (2.17) with respect to $z, x$ respectively and subtract. We obtain,

$$\rho_z(\psi_z\psi_{zx} - \psi_x\psi_{zz}) + \rho(\psi_z\psi_{zx} - \psi_x\psi_{zz})_z - \tag{2.18}$$

$$\beta\mu^2\rho_x(-\psi_z\psi_{xx} + \psi_x\psi_{xz}) -$$

$$\beta\mu^2\rho(-\psi_z\psi_{xx} + \psi_x\psi_{xz})_x = \rho_x \tag{2.19}$$

Using (2.14) and the fact that

$$\rho_x = \rho_\psi\psi_x, \quad \rho_z = \rho_\psi\psi_z, \tag{2.20}$$

we can eliminate $\rho$ from eq. (2.18) and obtain after some algebra

$$J\{\psi_{zz} + \mu^2\psi_{xx}, \psi\} - \tag{2.21}$$

$$N^2(\psi)J\{\frac{\beta}{2}(\psi_z^2 + \mu^2\psi_x^2), \psi\} = N^2J\{z, \psi\}$$

where

$$N^2(\psi) = -\frac{\rho_\psi}{\beta\rho} \tag{2.22}$$

is the nondimensional Brunt-Väisälä frequency which is (by definition) a function of $\psi$.

As a result we obtain the following equation for $\psi$ [Baines 1995,Nappo 2012].

$$\psi_{zz} + \mu^2\psi_{xx} - N^2(\psi)\left[z + \frac{\beta}{2}(\psi_z^2 + \mu^2\psi_x^2)\right] = G(\psi) \tag{2.23}$$

In (2.23), $G(\psi)$ is a function that has to be determined from the base flow. To do so we consider (2.23) at $x = -\infty$ and assume that the base flow is a function of $z$ only. Then express the left



hand side of (2.23) in terms of $\psi$ only to determine $G(\psi)$. (Here we assumed following [Yih 1967, Yih 1980, Baines 1995] that the disturbances from the base flow do not propagate upstream).

For example if we consider a shearless base flow with $u(-\infty, z) = 1$ then

$$\psi(-\infty, z) = z \tag{2.24}$$

and

$$G(\psi) = -N^2(\psi)(\frac{\beta}{2} + \psi). \tag{2.25}$$

Equation (2.23) becomes:

$$\psi_{zz} + \mu^2 \psi_{xx} - $$
$$N^2(\psi)\left[z - \psi + \frac{\beta}{2}\left(\psi_z^2 + \mu^2\psi_x^2 - 1\right)\right] = 0. \tag{2.26}$$

It follows from this example that different base flows at $x = -\infty$ will yield different functional forms of $G(\psi)$.

We consider now a perturbation $\eta$ from a shearless base flow $u(-\infty, z) = 1$ viz.

$$\eta = \psi - z. \tag{2.27}$$

Substituting this expression in (2.23) leads to

$$\eta_{zz} + \mu^2\eta_{xx} - \frac{N^2\beta}{2}(\eta_z^2 + \mu^2\eta_x^2 + 2\eta_z) + N^2\eta = 0. \tag{2.28}$$

## 2.2 Long's Equation with Free Convection

When the flow is not isothermal (2.4) has to be modified as follows

$$\rho(uw_x + ww_z) = -p_z - \gamma T\rho g \tag{2.29}$$

where $T$ is the temperature and $\gamma$ is the thermal expansion coefficient of the fluid. Moreover an equation for the temperature has to be added

$$\mathbf{u} \cdot \nabla T = \chi\nabla^2 T, \tag{2.30}$$





where $\chi$ is its thermometric conductivity. These equations hold under the assumption that

$$\frac{gh}{c^2} \ll \gamma T_0$$

where $h$ is the fluid column height, $c$ is the velocity of sound in the fluid and $T_0$ is the characteristic temperature difference.

We can non-dimensionalize these equations using (2.5) with the addition

$$\bar{T} = \frac{T}{T_0}$$

(as in the previous subsection we drop the bars). Eqs. (2.29), (2.30) become

$$\beta\rho(uw_x + ww_z) = -\mu^{-2}(p_z + \gamma T\rho) \tag{2.31}$$

$$\mathbf{u} \cdot \nabla T = \frac{1}{Pe}\nabla^2 T \tag{2.32}$$

where $Pe = \frac{U_0 L}{\chi}$ is the Peclet number.

Using (2.7) to introduce a stream function $\psi$, the momentum equations (2.9), (2.31) become

$$\beta\rho(\psi_z\psi_{zx} - \psi_x\psi_{zz}) = -p_x \tag{2.33}$$

$$\beta\rho(-\psi_z\psi_{xx} + \psi_x\psi_{xz}) = -\mu^{-2}(p_z + \gamma T\rho) \tag{2.34}$$

Using the same strategy as in the previous subsection to eliminate $p$ from these equations leads to

$$\rho_z(\psi_z\psi_{zx} - \psi_x\psi_{zz}) + \rho(\psi_z\psi_{zx} - \psi_x\psi_{zz})_z - \tag{2.35}$$
$$\mu^2\rho_x(-\psi_z\psi_{xx} + \psi_x\psi_{xz}) -$$
$$\mu^2\rho(-\psi_z\psi_{xx} + \psi_x\psi_{xz})_x = \frac{\gamma}{\beta}(T\rho)_x.$$

If the diffusion processes in (2.32) can be ignored i.e $|\frac{1}{Pe}\nabla^2 T| \ll 1$ then this equation can approximated by

$$J\{T, \psi\} = 0, \tag{2.36}$$



i.e. $T = T(\psi)$. Furthermore since $\rho = \rho(\psi)$ it follows that

$$(T\rho)_x = -J\{z, T\rho\} = -\frac{\partial(T\rho)}{\partial\psi}J\{z, \psi\}, \tag{2.37}$$

Using 2.14), (2.36) and (2.20) we can eliminate $\rho$ from eq. (2.35) and obtain after some algebra that

$$J\{\psi_{zz} + \mu^2\psi_{xx}, \psi\} - \tag{2.38}$$
$$N^2(\psi)J\{\frac{\beta}{2}(\psi_z^2 + \mu^2\psi_x^2), \psi\} = M^2 J\{z, \psi\}$$

where

$$M^2 = -\frac{\gamma}{\beta\rho}(T\rho)_\psi. \tag{2.39}$$

Using these definitions it follows that

$$\psi_{zz} + \mu^2\psi_{xx} - N^2(\psi)\frac{\beta}{2}(\psi_z^2 + \mu^2\psi_x^2) - M^2(\psi)z = G(\psi) \tag{2.40}$$

Eq. (2.40) can be considered as a "Generalized form of Long's equation". which include the effects of free convection. It contains two parameters $N^2$, $M^2$. The additional parameter $M^2$ controls the change of the temperature profile in the flow.

The function $G(\psi)$ in (2.40) can be determined using the same strategy as before. Thus if $\psi(-\infty, z)$ is given by (2.24) then

$$G(\psi) = -N^2(\psi)\frac{\beta}{2} - M^2(\psi)\psi \tag{2.41}$$

and eq. (2.40) becomes:

$$\psi_{zz} + \mu^2\psi_{xx} - N^2(\psi)\frac{\beta}{2}(\psi_z^2 + \mu^2\psi_x^2 - 1) - M(\psi)^2(z - \psi) = 0 \ . \tag{2.42}$$

For a perturbation $\eta = \psi - z$, from a base flow $u(-\infty, z) = 1$ we obtain from (2.40)

$$\eta_{zz} + \mu^2\eta_{xx} - \frac{N^2\beta}{2}(\eta_z^2 + \mu^2\eta_x^2 + 2\eta_z) + M^2\eta = 0 \tag{2.43}$$


## 2.3 Boundary Conditions and Approximations

We consider here numerical solutions of Long's equation over unbounded domain with a general base flow. The topography of the domain is represented by a function $h(x)$ whose maximum height is $H$. The boundary conditions that are imposed on the stream function $\psi$ are

$$\psi(-\infty, z) = \psi_0(z) \tag{2.44}$$

$$\psi(x, \tau h(x)) = \text{constant}, \quad \tau = \frac{H N_0}{U_0} \tag{2.45}$$

The constant in (2.45) which represents the value of the stream line over the topography $h(x)$ is (usually) set to zero.

To determine the proper boundary condition on $\psi(\infty, z)$ we note that Long's equation has no dissipation terms. Therefore radiation boundary conditions have to be imposed on $\psi$ in this limit. Similarly it is appropriate to impose radiation boundary conditions on $\psi(x, \infty)$ [Durran 1992].

When $|\tau| \ll 1$ the boundary condition (2.45) can be approximated (using (2.27) by

$$\eta(x, 0) = -\tau h(x). \tag{2.46}$$

When $N$, $M$ are set to a constant, (2.28), (2.43) become invariant with respect to translations in $x, z$. This implies that these equations admit self-similar solutions in the form $\eta = f(mx + nz)$ [Humi 2004]. These solutions represent gravity waves that are generated by the flow over the topography.

To compute numerical solutions for the perturbation $\eta$ over topography it has been common in the literature to consider (2.28) in the limit $\mu = 0$ and $\beta = 0$ [Durran 1992, Lily and Klemp 1979]. In addition $N$ is set to a constant or a step function over the computational domain.

In these limits (2.28) becomes a linear equation

$$\eta_{zz} + N^2 \eta = 0 \ . \tag{2.47}$$



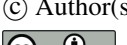

The limit $\beta = 0$ can be obtained either by letting $N_0 \to 0$ or $U_0 \to 0$. For the stratification to persists one has to assume that the limit $\beta = 0$ is obtained as $U_0 \to 0$.

Eq. (2.47) is a singular limit of (2.28). This is due to the fact that one of the leading second order derivatives drops when $\mu = 0$. Moreover the nonlinear terms in this equation drop out when $\beta = 0$. The approximate solutions that are derived from (2.47) and their physical limitations were considered extensively in the literature [Drazin and Moore 1967, Durran 1992, Humi 2004a, Humi 2006]. It was found that strong restrictions have to be imposed on the validity of these solutions even under the assumption that the base flow is shearless. However these approximations and the solutions that are derived from (2.47) are used routinely in the analysis of experimental atmospheric data [Shutts et al 1988, Baines 1995, Jumper et al 2004, Vernin et al 2007].

The general solution of eq. (2.47) is of the form

$$\eta(x, z) = q(x) \cos(Nz) + p(x) \sin(Nz). \tag{2.48}$$

The functions $p(x)$, $q(x)$ have to satisfy the boundary conditions derived from (2.45) and the radiation boundary conditions. To satisfy the radiation boundary conditions $p(x)$ and $q(x)$ have to satisfy [Baines 1995, Nappo 2012] that $p(x) = H[q(x)]$, where $H[q(x)]$ is the Hilbert transform of $q(x)$.

To satisfy the boundary condition on the terrain one has to solve the following integral equation [Drazin 1961, Lily and Klemp 1979, Durran 1992]

$$q(x) \cos(\tau N f(x)) + H[q(x)] \sin(\tau N f(x)) = -\tau h(x) . \tag{2.49}$$

# 3 Reductions and Transformations.

To begin with we observe that in (2.23), (2.40), (2.28), and (2.43) one can suppress the appearance of the parameter $\mu^2$ ($\mu \neq 0$) by applying the transformation $x = \mu \bar{x}$. Performing this transformation and assuming that $N$, $M$ are constants, these equations become invariant with respect to





translations in $x, z$. As a result they have solutions of the form $\eta = f(k\bar{x} + mz)$ [Humi 2004]. These are gravity waves that are generated by the atmospheric flow over the terrain.

Eq. (2.28) becomes

$$\eta_{zz} + \eta_{xx} - \alpha^2(\eta_z^2 + \eta_x^2 + 2\eta_z) + N^2\eta = 0. \tag{3.50}$$

where

$$\alpha^2 = \frac{N^2\beta}{2}$$

Similarly (2.43) becomes

$$\eta_{zz} + \eta_{xx} - \alpha^2(\eta_z^2 + \eta_x^2 + 2\eta_z) + M^2\eta = 0 \tag{3.51}$$

To these equations we apply the transformation

$$\phi = e^{-\alpha^2\eta} - 1. \tag{3.52}$$

Eqs. (3.50), (3.51) respectively become

$$\nabla^2\phi - 2\alpha^2\frac{\partial\phi}{\partial z} + N^2(1+\phi)\ln(1+\phi) = 0 \tag{3.53}$$

$$\nabla^2\phi - 2\alpha^2\frac{\partial\phi}{\partial z} + M^2(1+\phi)\ln(1+\phi) = 0 \tag{3.54}$$

Since $|\alpha^2\eta| \ll 1$ it follows that $|\phi| \ll 1$ and we can make the approximation $\ln(1+\phi) \approx \phi$. Equations (3.53) and (3.54) become

$$\nabla^2\phi - 2\alpha^2\frac{\partial\phi}{\partial z} + N^2(1+\phi)\phi = 0 \tag{3.55}$$

$$\nabla^2\phi - 2\alpha^2\frac{\partial\phi}{\partial z} + M^2(1+\phi)\phi = 0 \tag{3.56}$$

To simplify (3.55) and (3.56) we introduce the transformation

$$\phi = e^{\alpha^2 z}y. \tag{3.57}$$

Equation (3.55) becomes

$$\nabla^2 y + (N^2 - \alpha^4)y + N^2 e^{\alpha^2 z}y^2 = 0. \tag{3.58}$$





If $|\alpha^2 z| \ll 1$ (in domain of interest) we can approximate this equation by

$$\nabla^2 y + (N^2 - \alpha^4)y + N^2 y^2 = 0. \tag{3.59}$$

This equation has analytic closed form solution

$$y = \frac{3(N^2 - \alpha^4)}{n^2}\left[\tanh^2(C_1 + C_2 x - i\nu z) - 1\right] \tag{3.60}$$

where

$$\nu^2 = N^2 - \alpha^4 + 4C_2^2$$

and $C_1$, $C_2$ are integration constants.

Equation (3.60) represents solutions to a nonlinear equation for $y$ (and hence $\eta$). Since there is no superposition principle for these solutions, (3.60) represents therefore new "soliton type solution" for $\eta$ (in 3.50). Using the approximation $e^{\alpha^2 z} = 1 + \alpha^2 z$ this solution for $\phi$ (using (3.57)) satisfies (3.53) up to terms of order $\alpha^2$.

If $\alpha^2 z$ is not small one can approximate $e^{\alpha^2 z}$ by $1 + \alpha^2 z$ and use a perturbation expansion $y = y_0 + \alpha^2 y_1$ to compute $y_1$ (numerically).

Similar treatment can be applied to (3.56).

## 3.1 Linearized Equations and Solutions

To obtain a real solution for $\phi$ we neglect the $\phi^2$ term in (3.55) and (3.56) as being of second order. These approximations linearize (3.53) and (3.54) and yield (respectively)

$$\nabla^2 \phi - 2\alpha^2 \frac{\partial \phi}{\partial z} + N^2 \phi = 0 \tag{3.61}$$

$$\nabla^2 \phi - 2\alpha^2 \frac{\partial \phi}{\partial z} + M^2 \phi = 0 \tag{3.62}$$

These equations can be solved using separation of variables. Due to the similarity between (3.61) and (3.62) we discuss henceforth the solution procedure for (3.61) only.



If we substitute $\phi = f(x)g(z)$ in (3.61) and perform separation of variables we obtain the following equations for $f$, $g$

$$\frac{d^2f}{dx^2} + \omega^2 f = 0 \tag{3.63}$$

$$\frac{d^2g}{dx^2} - 2\alpha^2 \frac{dg}{dz} + (N^2 - \omega^2)g = 0 \tag{3.64}$$

Hence

$$f_\omega = A(\omega)e^{i\omega x} + B(\omega)e^{-i\omega x} \tag{3.65}$$

$$g_\omega = e^{\alpha^2 z}\left(C_1(\omega)e^{i\nu z} + C_2(\omega)e^{-i\nu z}\right) \tag{3.66}$$

where $C_1$, $C_2$, are constants and $\nu = \sqrt{N^2 - \alpha^4 - \omega^2}$. Hence for a wave to exist (in the z-direction) we must have $N^2 \geq \alpha^4 + \omega^2$. In addition the wave amplitude increases with height by a factor of $e^{\alpha^2 z}$.

Similarly for (3.62) we obtain the same expression for $f(x)$ and

$$g_\omega = e^{\alpha^2 z}(C_3(\omega)e^{i\lambda z} + C_4(\omega)e^{-i\lambda z}) \tag{3.67}$$

where $\lambda = \sqrt{M^2 - \alpha^4 - \omega^2}$.

The general solution of (3.61) can be written as

$$\phi = \tag{3.68}$$
$$e^{\alpha^2 z}\int [(D_1(\omega)e^{i(\nu z + \omega x)} + D_2(\omega)e^{-i(\nu z + \omega x)}]d\omega +$$
$$e^{\alpha^2 z}\int [D_3(\omega)e^{i(\nu z - \omega x)} + D_4(\omega)e^{-i(\nu z - \omega x)}]d\omega$$

Since the exponents multiplying $D_1$ and $D_2$ are conjugates it follows that for $\phi$ to be real we must have $\bar{D}_1 = D_2$ (where the bar stands for complex conjugation). Similarly we must have $\bar{D}_3 = D_4$.

The radiation boundary condition at $z \to \infty$ requires that the group velocity of the outgoing wave is positive. For a hydrostatic flow the dispersion relation is given by

$$\lambda(\omega) = \omega - \frac{sgn(\nu)N\omega}{\nu}$$




and the group velocity is

$$v_g = \frac{\partial \lambda}{\partial \nu} = \frac{sgn(\nu)N\omega}{\nu^2}$$

Hence $v_g > 0$ if $\nu\omega > 0$

Since the integration in (3.68) is over positive $\omega$ it follow then that the last two terms in this equation must be zero ($\nu\omega < 0$).

To satisfy the boundary condition (2.46) we observe (using (3.52)) that

$$\eta = -\frac{\ln(1+\phi)}{\alpha^2}. \tag{3.69}$$

Hence the boundary condition (2.46) becomes

$$\phi(x,0) = e^{\alpha^2\tau h(x)} - 1 \approx \alpha^2\tau h(x) \tag{3.70}$$

It follows then from (3.68) that

$$\int 2ReD_1(\omega)\cos(\omega x)d\omega \tag{3.71}$$
$$-\int 2ImD_1(\omega)\sin(\omega x)d\omega = \alpha^2\tau h(x)$$

This can be satisfied by standard Fourier integral expansion of $h(x)$.

The special case $\mu = 0$ was treated in detail in [Humi 2004] .

## 3.2  Application

To examine the application of the formulas derived above we consider the flow over a "witch of Agnesi" hill where the height of the topography is given by

$$h(x) = \frac{a^2}{(a^2 + x^2)}. \tag{3.72}$$

The Fourier integral expansion of $h(x)$ is

$$h(x) = \int_0^\infty A(\omega)\cos(\omega x)d\omega \tag{3.73}$$





where

$$A(\omega) = ae^{-a\omega}.$$

Using(3.71) this implies that $ImD_1 = 0$ and

$$D_1(\omega) = \frac{\alpha^2 \tau A(\omega)}{2}. \tag{3.74}$$

Substituting this result in (3.68) yields

$$\phi = e^{\alpha^2 z} \left\{ \int [D_1(\omega)e^{i(\nu z + \omega x)} + D_2(\omega)e^{-i(\nu z + \omega x)}]d\omega \right\}. \tag{3.75}$$

Hence,

$$\phi = \alpha^2 \tau e^{\alpha^2 z} \int e^{-a\omega} cos(\nu z + \omega x)d\omega \tag{3.76}$$

From this expression we can compute $\eta$ using (3.69). Fig. 1 displays the solution for $\eta$ for isothermal flow with $N = 1.5$, $\beta = 0.01$, $a = 1$, and $\tau = 1$. Fig. 2 displays the solution for $\eta$ for non-isothermal flow with the same parameters as in Fig. 1 but with $M = 2$.

# 4   Solutions with Shear

We consider here a base flow with $u = z$ i.e $\psi(-\infty, z) = z^2$. Using (2.23) to compute $G(\psi)$ we find that

$$G(\psi) = 2 - N^2(\psi^{1/2} + 2\beta\psi). \tag{4.77}$$

Long's equation (2.23) (with $\mu \neq 0$) becomes

$$\psi_{zz} + \mu^2 \psi_{xx} - N^2(\psi) \left[ z + \frac{\beta}{2}(\psi_z^2 + \mu^2 \psi_x^2) \right] = \tag{4.78}$$
$$2 - N^2(\psi^{1/2} + 2\beta\psi)$$

Applying the transformation $\bar{x} = \frac{x}{\mu}$ we obtain (after dropping the bars)

$$(\psi_{zz} - \alpha^2 \psi_z^2) + (\psi_{xx} - \alpha^2 \psi_x^2) - N^2 z = \tag{4.79}$$
$$2 - N^2(\psi^{1/2} + 2\beta\psi).$$



For a perturbation $\eta$ from the base flow i.e. $\psi = z^2 + \eta$ we obtain the following equation (where the square root was linearized assuming $|\eta| \ll 1$))

$$\eta_{zz} - 4\alpha^2 z\eta_z - \alpha^2(\eta_z)^2 + \eta_{xx} - \tag{4.80}$$
$$\alpha^2(\eta_x)^2 + \left(4\alpha^2 + \frac{N^2}{2z}\right)\eta = 0.$$

We introduce now the transformation

$$\phi = e^{-\alpha^2\eta} - b \tag{4.81}$$

where $b \neq 0$ is a parameter to be determined latter. Applying this transformation to (4.80) and making the approximation $\ln(b + \phi) = \ln(b) + \frac{\phi}{b}$ (assuming $|\phi| \ll b$) leads to the following

$$2bz\phi_{zz} + 2bz\phi_{xx} - 8b\alpha^2 z^2\phi_z + \tag{4.82}$$
$$(8\alpha^2 z + N^2)[\phi^2 + b(\ln(b) + 1)\phi + b^2\ln(b)] = 0.$$

Dropping the nonlinear term in $\phi^2$ and letting $b = e^{-1}$ (to suppress the term containing $\phi$) (4.82) becomes

$$2z\phi_{zz} + 2z\phi_{xx} - 8\alpha^2 z^2\phi_z - e^{-1}(8\alpha^2 z + N^2) = 0 \tag{4.83}$$

A particular solution $\phi_p$ of this (linear) equation is [Abramowitz and Stegun 1974]

$$\phi_p = -\frac{1}{4}\int e^{2\alpha^2 z^2 - 1}[-4\alpha\sqrt{2\pi}\,erf(\sqrt{2}\alpha z) + \tag{4.84}$$
$$N^2\Gamma(0, 2\alpha^2 z^2)]dz$$

The homogeneous part of (4.83) can be solved by separation of variables viz. $\phi = f(x)g(z)$ where $f(x)$ satisfies (3.63). The resulting equation for $g(z)$ has analytic solution in terms of Kummer functions [Abramowitz M. and Stegun 1974].

$$g(z) = C_1 zKummerM(\nu_1, \frac{3}{2}, 2\alpha^2 z^2) + \tag{4.85}$$
$$C_2 zKummerU(\nu_1, \frac{3}{2}, 2\alpha^2 z^2)$$

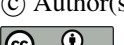

**Nonlinear Processes**
**in Geophysics**
Discussions



where $\nu_1 = \frac{4\alpha^2 + \omega^2}{8\alpha^2}$.

For $\mu = 0$ the equation for the perturbation $\eta$ is

$$\eta_{zz} - 4\alpha^2 z \eta_z - \alpha^2 (\eta_z)^2 + \eta \left( \frac{N^2}{z} + 4\alpha^2 \right) = 0. \tag{4.86}$$

Applying the transformation (4.81) to (4.86) with $b = e^{-1}$ and omitting the nonlinear term in $\phi^2$ we obtain for $\phi$ the same equation as (4.83) without the derivatives with respect to x. A particular solution of this equation is given by (4.84) while the solution of the homogeneous equation is

$$\phi(z) = c_1 \, erf(i\sqrt{2}\alpha z) + c_2 \tag{4.87}$$

where $c_1$, $c_2$ are constants.

# 5    Summary and Conclusions.

Computing numerical solutions for Long's equation has been always a challenge even in some (singular) limiting cases. In this paper we introduced a transformation of this equation which under mathematically acceptable approximations leads to analytic expressions for the solutions. In particular these solutions capture the dependence of the wave amplitude on the height.

The paper provides also an extension of Long's equation to the case where the atmospheric flow is not isothermal. This new equation can be solved analytically by the same transformation that is used for Long's equation.

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





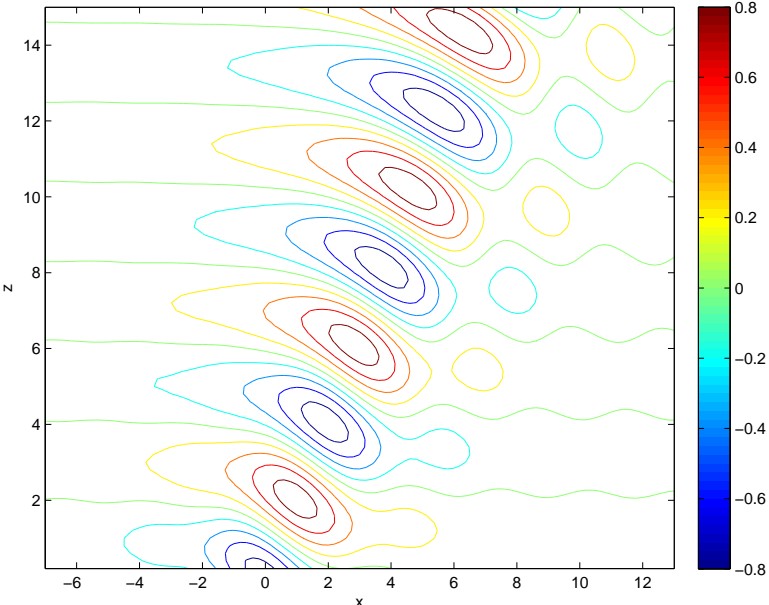

Figure 1: Contour plot of $\eta$ for isothermal flow over a topography





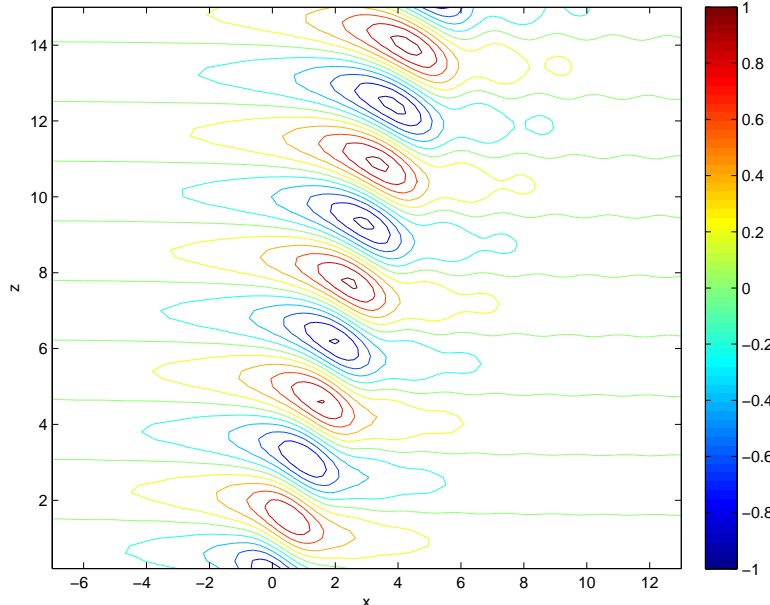

Figure 2: Contour plot of $\eta$ for non-isothermal flow over a topography