# Peer review of "Analytic Solutions for Long's Equation and its Generalization"

_Nonlinear Processes in Geophysics, 2017_

## Referee Comment (RC1) · Anonymous Referee #1 · 7 Aug 2017

This paper is one of a series by this author on various forms of "Long's equation". Longs equation is a linear form of equation for steady density-stratified flow over an obstacle, and depends on a particular form of upstream flow, generally uniform with height, or approximately so. The solutions of it are limited to obstacles of sufficiently small height because, if the obstacle is tall enough, experiments show that it generates upstream-propagating disturbances of a columnar nature that effectively alter the upstream velocity and density profiles. The author's publications on this topic do not seem to address this issue and the resulting limitations on applicability of his solutions, though in section 2.3 of this paper it appears that a variety of upstream conditions can be chosen (2.44).

The author does introduce a variety of factors into his equations (here, additional "temperature" variations), and the solutions shown in Figures 1 and 2 look quite realistic, but the analysis appears to be effectively a form of steady-state linear analysis. The main virtue of this paper is that the author gives analytic expressions for the solutions, for uniform upstream flow in section 3.2, and for uniformly sheared upstream flow in section 4 (though there are no figures for the latter).

—————————————————————————

---

## Author Comment (AC1) · 8 Aug 2017

[12pt]article amsmath

**Reply to Anonymous Referee #1**

August 8, 2017

1. With all due respect to the referee the claim that

"Longs equation is a linear form of equation..." is incorrect.

The classical form of Long's (equation 2.23 in the paper) for the stream function $\psi$ and the perturbation $\eta$ (eq. 2.28) are BOTH NONLINEAR equations.

In the literature some authors IGNORE the nonlinear terms to obtain eq. 2.47 . However this is a singular limit of the equation and some of the "physical contents" of the solution is lost.(e.g the dependence of $\psi$ on the height).

2. Solutions of Long equation are used routinely in the experimental analysis of gravity waves. Therefore one can not underestimate the practical importance of this equation (see my bibliography for references). (Many topographical obstacles are of moderate heights)

3. The referee seems to be unaware of my previous paper: "Time Dependent Long's Equation, Nonlin. Processes Geophys., 22, pp. 133-138 (2015)". In which I offered an extension of Long's equation to time dependent flows.

4. The NOVEL part of the current paper consists of a sequence of transformations

which "somehow" linearize Long's equation and lead to analytic form of the solution WITHOUT scarifying any of the physical contents of the equation. In the process we find also "solitonic form solutions" of this equation which never appeared in the literature before. The solutions presented also show how the amplitude of the gravity waves depend on the height.

5. The statement that Long's equation requires "particular form of the upstream flow" is incorrect. As was shown in the paper e.g eq 2.44 (as the referee admits).

---

## Referee Comment (RC2) · Anonymous Referee #2 · 10 Sep 2017

The novel aspects of this paper are:

(i) Makes a (slight) generalization of the classical Long's equation to non-isothermal flows

(ii) Derives analytic solutions under certain (special) conditions

Both these aspects could be useful if the author discussed the appropriate physics in some detail. Particularly in regard to (ii), the new solutions are obtained via a sequence of unmotivated transformations, and the final results shown in Fig 1 and 2 are not explained at all (eg, what is the effect of nonlinearity, etc)

Also, a couple of side remarks:

(i) Long's equation was first obtained by Dubreil-Jacotin (1935)

(ii) A major assumption of Long's equation, not mentioned by the author, is restriction to 2D; this is discussed in Yih (1967) and Akylas & Davis (2001)
* * *

---

## Author Comment (AC2) · 20 Sep 2017

[12pt]article amsmath

[Figure]

**Reply to Anonymous Referee #2**

September 20, 2017

1. The transformations on Long's equation which are introduced in this paper were "inspired" by the transformations which are used to linearize Ricatti and Burger's equations (log-type transformation). From a physical point of view the motivation comes from the desire to replace the nonlinearities due to the derivatives of $\eta$ in (**??**) by expressions that correspond to $\eta$ itself. This replacement will enable us to make approximations which are based on physical insights.

2. As to the figures: They demonstrate the change in the gravity wave amplitude with height and the effect that non-isothermal flow can have on this wave.

I revised the paper to reflect these changes (in red)

**Side Remarks**

1. I added a reference to Dubreil-Jacotin paper.

2. Although the restriction to 2D is obvious I added a remark to this effect in the introduction and added a reference to the paper by Akylas & Davis (2001) (The reference to Yih is already there).

---

## Author Response (AR2)

**Reply to the Editor and Anonymous Referees**

October 9, 2017

**1 Reply to the Editor**

Dear Prof. Fernando, Editor NPG

There must have been some miscommunication about my replies to the referees. I did reply to them point by point (see below) but did not emphasize enough the novel aspects of the paper. In view of your kind remarks I expanded my reply to the first referee in item #4 below. (The expansion is in magenta color and boldface). I emphasized these novel aspects of my results in the revised paper which is being submitted through the NPG-portal.

Thank you for your kind help, Sincerely, Mayer Humi.

**2 Reply to Anonymous Referee #1**

The following are my point by point replies to the first referee:

1. With all due respect to the referee, his claim that

   "Longs equation is a linear form of equation..." is incorrect.

   The classical form of Long's (equation 2.23 in the paper) for the stream function $\psi$ and the perturbation $\eta$ (eq. 2.28) are BOTH NONLINEAR equations.

   In the literature some authors approximate the equation by IGNORING the nonlinear terms in order to be able to solve eq. 2.28 **numerically**. This leads to eq. 2.47 . However this is a singular limit of Long's equation and as a result some important "physical aspects" of the solution are lost by this approximation (e.g the dependence of $\psi$ on the height).

2. Solutions of Long equation are used routinely in the experimental analysis of gravity waves. Therefore one can not underestimate the practical importance of this equation (see my bibliography for references). (Many topographical obstacles are of moderate heights)

3. The referee seems to be unaware of my previous paper: "Time Dependent Long's Equation, Nonlin. Processes Geophys., 22, pp. 133-138 (2015)". In which I offered an extension of Long's equation to time dependent flows.

4. The NOVEL part of the current paper consists of a sequence of transformations which "somehow" linearize Long's equation and lead to analytic form of the solution WITHOUT scarifying any of the physical contents of the equation. In the process we found also NEW "solitonic form solutions" of this equation which **never** appeared in the literature before. The solutions presented also show how the amplitude of the gravity waves depend on the height. In addition the paper presents an extension of Long's equation to non-isothermal flows. The presentations in subs-sections 2.1 and 2.3 are needed in order to put the new novel aspects of this paper in context and give the reader a sense of their importance. **The bulk of the paper which comprise of subsection 2.2 and sections** 3, 4 **presents completely NEW results which NEVER appeared before**.

5. The statement made by the referee that Long's equation requires "particular form of the upstream flow" is incorrect. As was shown in the paper e.g eq 2.44 (as the referee admits).

**3   Reply to Anonymous Referee #2**

The following are my point by point replies to the second referee:

1. The transformations on Long's equation which are introduced in this paper were "inspired" by the transformations which are used to linearize Ricatti and Burger's equations (log-type transformation). From a physical point of view the motivation comes from the desire to replace the nonlinearities due to the derivatives of $\eta$ in eq 3.50 (or 3.51) by expressions that correspond to $\eta$ itself. This replacement will enable us to make approximations which are based on physical insights.

2. As to the figures: They demonstrate the change in the gravity wave amplitude with height and the effect that non-isothermal flow can have on this wave.

**Referee Side Remarks**

1. I added a reference to Dubreil-Jacotin paper.

2. Although the restriction to 2D is obvious I added a remark to this effect in the introduction and added a reference to the paper by Akylas & Davis (2001) (The reference to Yih is already there).

I revised the paper to reflect **all** these changes in red.